# A Placebo-Controlled Trial to Evaluate Two Locally Delivered Antibiotic Gels (Piperacillin Plus Tazobactam vs. Doxycycline) in Stage III–IV Periodontitis Patients

**DOI:** 10.3390/medicina59020303

**Published:** 2023-02-07

**Authors:** Ioana Ilyes, Darian Rusu, Viorelia Rădulescu, Octavia Vela, Marius Ion Boariu, Alexandra Roman, Petra Surlin, Giorgios Kardaras, Simina Boia, Salvatore Chinnici, Holger Friedrich Rudolf Jentsch, Stefan-Ioan Stratul

**Affiliations:** 1Department of Periodontology, Faculty of Dental Medicine, Anton Sculean Research Center for Periodontal and Peri-Implant Diseases, Victor Babes University of Medicine and Pharmacy, 300041 Timisoara, Romania; 2Department of Endodontics, Faculty of Dental Medicine, TADERP Research Center, Victor Babes University of Medicine and Pharmacy, 300041 Timisoara, Romania; 3Department of Periodontology, Faculty of Dental Medicine, Applicative Periodontal Regeneration Research Unit, Iuliu Hatieganu University of Medicine and Pharmacy, 400347 Cluj-Napoca, Romania; 4Department of Periodontology, Faculty of Dental Medicine, University of Medicine and Pharmacy, 200349 Craiova, Romania; 5Department of Cariology, Endodontology and Periodontology, Centre for Periodontology, University Hospital of Leipzig, 04103 Leipzig, Germany

**Keywords:** clinical trials, subgingival instrumenation, antimicrobial therapy, local drug delivery, local antibiotics, piperacillin plus tazobactam, doxycycline

## Abstract

*Background and objectives*: this study aims to evaluate the clinical and microbiological effects of a single subgingival administration of a locally delivered antibiotic gel containing piperacillin plus tazobactam and compare it with a slow-release doxycycline (14%) gel and a placebo gel, following subgingival instrumentation (SI) in patients with severe periodontitis. *Materials and methods:* sixty-four patients diagnosed with stage III–IV periodontitis were enrolled, were randomly assigned into three groups, and were treated additionally with a single subgingival administration of piperacillin plus tazobactam gel (group A); doxycycline gel (group B); and placebo gel (group C). The primary outcome variable was the change in mean probing pocket depth (PPD) 6 months after the intervention. Secondary outcome variables were changes in mean full-mouth bleeding score (FMBS); full-mouth plaque score (FMPS); overall bleeding index (BOP); pocket closure; and clinical attachment level (CAL), along with changes in the numbers of five keystone bacteria: *Aggregatibacter actinomycetemcomitans* (*A.a.*), *Porphyromonas gingivalis* (*P.g.*)*, Prevotella intermedia* (*P.i.*)*, Tannerella forsythia* (*T.f.*), and *Treponema denticola* (*T.d.*). Intergroup and intragroup differences were evaluated at 3 and 6 months. *Results:* at baseline, the three groups were comparable. An improvement in clinical parameters such as PPD, CAL, and BOP between groups was observed at 3 and 6 months, but without statistical significance (*p* > 0.05). At 6 months, the intragroup analysis showed a significant reduction in clinical parameters. Even though the piperacillin plus tazobactam group showed slightly higher PPD reduction, this was not statistically significant when compared to both control groups. *Conclusions:* The groups had similar results, and subgingival instrumentation can be executed without adjunctive antimicrobials, reducing the costs for the patient and the working time/load of the professional.

## 1. Introduction

The presence of bacterial plaque represents the principal etiologic factor involved in the initiation and progression of inflammatory periodontal diseases [1,2]. The goal of nonsurgical periodontal therapy is the elimination of the suspected bacterial pathogen(s) to arrest the destruction of the periodontium [3]. Mechanical therapy, such as scaling and root planing, is the main nonsurgical periodontal treatment (SRP).

Microbial plaque is a major contributor to periodontitis. If patients with periodontal disease maintain good oral hygiene after scaling and root planing (SRP), they experience a reduction in inflammation, a decrease in periodontal probing depth (PPD), and an increase in clinical attachment level (CAL) [4,5]. An essential objective during the treatment of periodontitis is to achieve a stable attachment of the gingival tissue to the tooth, by removing the subgingival bacterial plaque and subsequently preventing microbial recolonization of the periodontal pockets using effective plaque control measures. To achieve this goal, there are a couple of approaches: mechanical therapy with or without flap elevation [6,7] with or without the adjunctive use of local (antiseptics or antibiotics) or systemic (antibiotics) antimicrobial agents [8,9,10,11,12,13].

SRP, more recently renamed as subgingival instrumentation (SI) (by The EFP S3 level Clinical Practice Guideline (European Federation of Periodontology)) [14], is considered to be the gold standard in the nonsurgical treatment of periodontitis [15,16,17,18,19]. Several factors may influence its effectiveness and success, including the presence of deep periodontal pockets [20,21] and furcation damage [22,23,24]. SI has its limitations, and in some individuals (for example, those with grade C periodontitis) [25] or at certain sites, it may not have the intended effects. In such cases, the use of local antibiotic agents as an adjunct to SI may prove to be beneficial. Combining traditional SI with intrapocket antibiotic application is one of the potential alternative procedural protocols that allows for the acquisition of long-term minimum concentration of the drug inhibiting periopathogen development [16].

Antibiotic substances administered locally or systemically are the most effective chemical agents. However, due to the organization of biofilm-based bacteria, antibiotic efficacy is limited [26,27]. The advantages of using local antibiotic placement compared to systemic administration include minimal systemic load, better patient compliance, effective placement strength over several days, and improved pharmacokinetic response [28]. These drugs are used in periodontal pockets and can inhibit or eliminate periodontopathogenic microorganisms as well as modulate the inflammatory response of the tissues [29]. Locally delivered antimicrobials are created with the relevant medications impregnated in a carrier and are offered as gels, fibers, chips, polymers, or ointments.

The main advantages of the local treatment are fewer side effects and improved compliance, in comparison with drugs used systemically, and reduced chances of developing bacterial tolerance to medications [30]. Multiple studies and an array of systematic reviews have assessed the effects of local antimicrobials delivered in fibers, gels, chips, or microspheres, mainly in untreated patients but also in treated sites with poor response or with recurrent disease [31,32,33,34]. Evidence indicates that local delivery of antimicrobials in periodontal pockets as an adjunct to SI and as aids in the control of the growth of pathogenic bacteria provides an adjunctive benefit compared to SI alone [32]; local drug delivery (LDD) systems provide a higher concentration of the antimicrobial with a sustained release over a longer duration of time [35].

LDD must fulfill the following criteria to be clinically effective: 1. it must reach the intended site of action; 2. it must remain at an effective concentration at the adequate site of action; 3. it should last for a sufficiently adequate duration of time [28,29,36]. To complement the nonsurgical periodontal therapy, multiple options of antimicrobials that can be locally delivered have been proposed, such as metronidazole, chlorhexidine, minocycline, doxycycline, and tetracycline [37]. However, the effectiveness of topical periodontal antibiotic therapy has not been conclusively proven, despite being vigorously marketed, and few products are still currently in use. Over the last decade, more antimicrobials have been added every year to the current inventory of LDDs for periodontal use, with various outcomes, but few are being retained by the clinical practice.

Recently, a formulation of LDD for intrapocket application, based on a combination of beta lactam/beta lactamase inhibitor, has been introduced in the periodontal practice. It is a patented mixture that consists of piperacillin plus tazobactam, combined with a carrier. After mixing, the preparation gellifies and forms a coating that seals the periodontal pocket. These characteristics should help the product’s active ingredients stay where they were intended—in the area where the preparation was applied. According to the manufacturer, over the course of 8 to 10 days, the components of Gelcide^®^ are gradually released at quantities that are higher than the minimal inhibitory concentration. The coating is permeable, but it is insoluble to fluid, protecting the pocket from exposure to bacteria in the oral cavity and from further irritation and infection [16].

The objective of this randomized, controlled, double-blind study was to evaluate clinically and microbiologically the efficacy of the combination of piperacillin plus tazobactam vs. doxycycline (slow-released doxycycline), both in the form of locally delivered gels in periodontal pockets, as an adjunct to SI, compared to a placebo gel adjunctive to SI, in stage III–IV periodontitis.

## 2. Materials and Methods

### 2.1. Study Design

This study was conducted as a double-blinded, randomized, placebo-controlled clinical trial of 6 months with a parallel design of three independent groups by a 1:1:1 allocation ratio.

It was established that 19 patients per group are needed to detect a significant mean difference of 1 mm in PPD reduction between groups, assuming a common standard deviation of 1 mm, 80% power, and a significance level of 0.05. The Pitman asymptotic relative efficiency correction was used in the sample size calculations to account for the use of nonparametric tests. Considering an anticipated drop-out rate of ~10%, it was decided to enroll at least 21 patients in each group.

This study included a cohort of 100 patients equally divided into one of the three groups: A, B, and C. Subjects were selected from patients of the Department of Periodontology of the “Victor Babeș” University of Medicine and Pharmacy Timisoara, Romania. The study design was approved by the Committee for Research Ethics of the Victor Babes University of Medicine and Pharmacy Timisoara (approval No. 20/03.09.2018) and conforms to the requirements of the Declaration of Helsinki as adopted by the 18th World Medical Assembly in 1964 and subsequently revised. All subjects were informed about the nature and purpose of this study, and each subject signed an informed consent document giving permission for the dental procedures and sampling of biological material. This study was carried out between October 2018 and February 2021. This study is registered in the ISRCTN89207035 Registry of Clinical Trials and follows the guidelines described in the CONSORT 2010 statement on clinical trials.

Diagnosis (staging and grading) of periodontitis was based on the criteria outlined in The New Classification Scheme from 2018 [25]. Furcation involvement was classified according to Hamp [38]. Prior to this study, the examiner (specialist of Periodontology) was calibrated, the intraexaminer calibration for reliability testing resulted in κ = 0.92 for repeated measurements of PPD and CAL in two quadrants of five patients, other than the patients recruited for this study (to complete the evaluations needed for this study in a reliable and accurate manner that is consistent with current standards for clinical periodontal studies [39]). The medical history was completed and following were considered: possible systemic disorders, smoker status, and compliance. Each patient was repeatedly trained on oral hygiene measures, and the plaque and calculus were removed by supragingival scaling one week before the initial evaluation.

All subjects were diagnosed with periodontitis stage III and IV. All participants fulfilled the following inclusion criteria: subjects aged over 25 years, at least 8 sites with PD ≥ 5 mm and showing bleeding on probing, clinical attachment loss ≥ 5 mm, patients who have not undergone periodontal therapy within the last 12 months.

Patients with the following conditions were excluded: clinically relevant psychiatric disorders, alcohol consumption, autoimmune disorders, HIV infection, untreated diabetes mellitus, pregnancy or breastfeeding, patients who have received periodontal therapy in the last 12 months, patients who reported local and/or systemic antibiotic therapy within the 3 months before the baseline examination of this study, candidiasis, allergies to piperacillin, tazobactam, doxycycline, or to any tetracycline or penicillin or to any excipient of the products used, systemic medication that can influence the clinical features of periodontitis, patients who have rinsed or irrigated with antiseptics less than a month before the initial examination, conditions that require antibiotic protection.

### 2.2. Clinical Examination

The following parameters were measured in the initial evaluation and at 3 and 6 months: Overall Plaque Index (PlI) in 6 sites per tooth; Probing Pocket Depth (PPD) evaluation on the vestibular, palatal, and lingual surfaces was performed halfway between the line angles using the PCP–UNC 15 periodontal probe (Hu–Friedy, Chicago, IL, USA). On the interproximal faces, the evaluation was performed in the immediate vicinity of the contact point with optimal pressure *(20–25 N)*; Overall Bleeding Index (BOP) was evaluated in 6 sites per tooth at 30 s from the probing; level of clinical attachment (CAL) [40]; teeth mobility degrees [41]; involvement of furcation (FI) [38] in multi-rooted teeth, all sites with furcation involvement were included into grade I, II or III; all clinical measurements were performed by one clinician (I.I.) using a Nabers probe. Data were recorded in the periodontal sheet of the University of Bern (http://www.periodontalchart-online.com/uk/index.asp, accessed 1 October 2018), were saved in PDF format, were printed and included in the observation file of each patient. Patients fulfilling the inclusion criteria received an informed agreement that he/she had 7 days to analyze it and had to sign it to be included in this study. During the study period, the attending examiner monitored the patients for disease progression. The patients were excluded from this study in case of progressive attachment loss of 2 mm or more between two subsequent evaluation time points, as re-instrumentation of the affected sites was deemed necessary. If that occurred, the patients received treatment as necessary.

### 2.3. Microbiological Examination

Microbiological samples were obtained from sites with a depth of minimum 5 mm at the initial examination, with 4 sites being selected, 1 in each dial. These sites were used as reference sites for samples collected at 0 and 6 months. The subgingival plaque was sampled for microbiological evaluation as follows: the site was isolated with rolled wool, the overgrowth plaque was removed with a sterile compress, and the gingival surface was dried, and plaque samples were obtained by inserting 2 sterile ISO #30 paper cones into the site, which were left in place for 30 s for saturation [42]. Plaque samples were obtained at baseline (prior to treatment of the patient) and at 6 months after the initial evaluation. The paper points were pooled immediately into sterile sealed Eppendorf tubes and sent for polymerase chain reaction (PCR). The PCR testing was conducted at the laboratories of the Department of Biochemistry of the “Victor Babeş” University of Medicine and Pharmacy.

Detection of the major periodontopathogens *Aggregatibacter actinomycetemcomitans (A.a.)*, *Porphyromonas gingivalis (P.g.)*, *Prevotella intermedia (P.i.)*, *Tannerella forsythia (T.f.)*, and *Treponema denticola (T.d.)* was carried out by molecular genetic analysis of the samples taken. Presence of these bacteria was assessed using a commercial kit micro-IDent^®^ (Hain Lifescience, Nehren, Germany). The same sites were used to collect the microbiological samples during the 6-month reevaluation time point.

### 2.4. Randomization

The team that took part in this study was composed of an examiner (specialist of Periodontology, I.I.), a randomizer (a person performing the computerized randomization who was not necessarily a physician), and an operator (specialist of Periodontology, V.R.). Randomization was performed using a number generator (www.randomizer.org, accessed 1 October 2018) by an independent individual (randomizer). This site was used to generate 3 sets of numbers with 33 numbers per set, and the numbers ranged from 1 to 99; each number was unique, and these numbers were ordered in ascending order. Each patient accepted in this study was assigned a number corresponding to a study group. Due to the particular physical nature and package of the products, only the patients and the evaluator were blinded. Blinding of the patients was literally achieved by covering the patients’ eyes during the gel application. The differences between the original packages of the gels and the obvious differences in consistency and color precluded the blinding of the operator.

### 2.5. Treatment Procedures

Each patient was assigned to one of the three treatment groups according to computer-generated randomization: group Gelcide^®^ (A): ultrasonic SI using EMS Piezon^®^ Master and PerioSlim inserts (EMS, Nyon, Switzerland) and manual SI using Gracey curettes (Hu–Friedy #5/6; 7/8; 11/12; 13/14); application of Gelcide^®^ (Italmed, Firenze, Italy) according to manufacturer’s instructions. group Ligosan^®^ (B): ultrasonic SI and manual SI using Gracey curettes (Hu–Friedy #5/6; 7/8; 11/12; 13/14); application of Ligosan^®^ (Kulzer GmbH, Hanau, Germany) according to manufacturer’s instructions. Group CONTROL (C): ultrasonic SI and manual SI using Gracey curettes (Hu–Friedy #5/6; 7/8; 11/12; 13/14); application of placebo gel.

Periodontal treatment was performed, including oral hygiene instruction and supragingival ultrasonic instrumentation, by the person designated as the operator. Standard oral health instructions were recommended: tooth brushing, either with manual or powered toothbrushes, minimum 2 min twice per day, interdental cleaning with interdental brushes. Instructions were personalized according to the patient’s need to obtain the best plaque control. No antiseptics were recommended. Subgingival plaque samples were collected from 4 tooth sites at baseline and 6 months (after supragingival ultrasonic instrumentation, at 7 days). Supra and subgingival SI by ultrasound and manual instrumentation were performed at all sites: ultrasound instruments in 10 min, followed by manual SI with Gracey curettes; #5/6; 7/8; 11/12; 13/14; instrumentation was followed by subgingival gel application, depending on the patient’s group.

For patients assigned to group A (test), GELCIDE^®^ was applied according to the manufacturer’s recommendations. First, the solution was injected into the powder container, and the container was shaken until the solution became homogeneous. The necessary quantity was then extracted from the mixed container with a syringe and inserted into the periodontal pocket at the apical extremity of the pocket. Once the gel has been applied, the excess is removed using a curette and a cotton ball. After application, a gentle air jet was applied. The patients assigned to group B (positive control) received LIGOSAN^®^. The product was transported by plastic cannula, and the gel application was performed with a syringe inside the oral, vestibular, medial, and distal pockets, and at the apical extremity of the pocket. Once the gel had been applied, the supragingival excess was removed using the curette and a cotton ball. After application, a gentle air jet was applied. For the patients assigned to group C (placebo), a placebo gel was applied into the periodontal pocket in the most apical portion.

Patients from all groups were instructed to gently brush and refrain from flossing for the first 36 h following treatment in treated area. After 36 h, oral hygiene procedures were resumed: gentle brushing of the application area was performed twice a day and removal of interdental plaque once a day. The timeline of the treatment is represented in Figure 1.

### 2.6. Statistical Analysis

The main outcome variable was the reduction in PPD at 3 and 6 months after intervention, while BOP, CAL, REC, FMPS, FMBS, pocket closure, and detection scores of the five selected bacterial species’ changes (*A. actinomycetemcomitans*, *P. gingivalis*, *T. forsythia*, *P. intermedia* and *T. denticola*) were regarded as secondary outcomes. For each of the quantitative variables PPD, REC, CAL, a patient mean value was computed per time point, which was further used in the statistical analyses. Differences between groups for variables measured on a continuous or ordinal scale were analyzed using Kruskal–Wallis tests, with post hoc Mann–Whitney pairwise tests as necessary. Proportions were compared by chi-square tests. Assessment of intragroup differences between successive time points for quantitative variables was performed using Friedman tests, with subsequent Wilcoxon signed-rank tests for pairwise comparisons. The Bonferroni correction was used to account for multiple comparisons. *p* values < 0.05 were accepted for statistical significance. The statistical analyses were performed using the software R version 4.1.2 [43]. Changes in the detection frequency scores of the main keystone bacteria were evaluated in terms of the microbiological status. Results were noted and categorized into one of four groups: 0 = nondetectable, 1 = detectable < 10^4^ (10^3^ for *A.a.*), 2 = 10^4^–10^5^ (10^3^–10^4^ for *A.a.*), 3 = 10^5^–10^6^ (10^4^–10^5^ for *A.a.*), and 4 ≥ 10^7^ (10^6^ for *A.a.*) [42]. Using the Wilcoxon signed-rank test, intragroup comparisons of detection scores of pathogen species between the baseline and 6-month reevaluation time points were made. For intergroup comparisons of the detection scores at each time point, the Kruskal–Wallis test was applied.

## 3. Results

No side effects or negative impacts such as pain or dentinal hypersensitivity directly connected to the applied gels were reported.

Figure 2 shows the CONSORT flow diagram for this study. Over the 6-month follow-up, one patient withdrew his consent after the treatment assignment. Because he did not receive any treatment, he was excluded from all efficacy data sets. A total of 36 patients (36%) of the 100 treated patients discontinued this study prematurely and were excluded by the investigator: 5 patients due to attachment loss ≥ 2 mm and need for other treatment, and 16 patients needed systemic antibiotic therapy for different systemic conditions. The remaining 14 patients were assigned to treatment but discontinued, with no-show behavior during the COVID-19 pandemic, and were lost to follow-up. Data were studied in 64 subjects that completed this study.

### 3.1. Clinical Results

Table 1 presents the characteristics of the patients at baseline (demographic data), full-mouth clinical parameters at baseline, and follow-up visits in the three groups. The mean age of the study population was 50.71 years (SD ± 9.56) in group A, 47.32 years (SD ± 8.08) in group B, and 49.95 years (SD ± 6.61) in group C, and there were no statistically significant differences between control and test groups in terms of demographics (differences regarding sex, smoking, age) or clinical presentation of periodontitis. The number and frequency distribution of sites with different baseline PPD in the test and control groups show the number and prevalence of sites that received treatment, grouped according to different baseline PPD. PPD measurements demonstrated no significant differences between study groups at baseline. The intragroup distribution was well pondered; a total of 228 reference sites were analyzed. The PPD of the sites ranged from 5 to 8 mm at baseline. The mean PPD at baseline was 5.89 ± 0.58 mm for the Gelcide^®^ group, 6.03 ± 0.58 mm for the Ligosan^®^ group, and 5.89 ± 0.63 mm for the placebo group (Table 1). PPD measurements demonstrated no significant differences between study groups at baseline, 3 or 6 months. A statistically significant difference for PPD in every study group was present. In each group, there are statistically significant differences between PPD recorded at the different time points (Friedman tests, *p* < 10^−7^ in each case); post hoc Wilcoxon tests indicate significant differences between baseline and 3 months, baseline and 6 months, as well as 3 and 6 months (see Figure 3).

The evolution of PPD follows a similar dynamic in the three groups: a pronounced decrease after the first 3 months, followed by a slight increase at 6 months. However, the groups had similar results.

Regarding the variation of REC, there were no statistically significant differences between groups at any of the analyzed time points; the Friedman tests showed in each group significant differences between RECs measured at different time points (*p* < 10^−5^ in each case). More precisely, in each group, the REC at baseline is significantly lower than at 3 and 6 months (*p* < 0.01 for all the respective pairwise comparisons performed), and the values recorded at 3 and 6 months are similar (see Figure 4).

With regard to changes in CAL, there were no statistically significant differences between groups at any of the analyzed time points, with similar CAL dynamics in the three groups: significant differences in the values recorded at different time points (Friedman tests, *p* < 10^−7^ for all intragroup comparisons); Wilcoxon post hoc tests show differences in each group between baseline and 3 months, baseline and 6 months, and 3 and 6 months (*p* < 0.001 in each case). More precisely, there is a significant decrease in CAL between baseline and 3 months. An improvement occurred in all three groups compared to baseline, followed by an increase at 6 months (see Figure 5).

Comparisons of FMPS measurements show slight differences between the groups at 3 months; however, subsequent pairwise comparisons using Mann–Whitney tests do not indicate these differences to be statistically significant (*p* > 0.05 in all cases, after applying the correction for multiple comparisons). Slight differences existed between groups at baseline, at 6 months, and regarding FMBS improvement at 6 months compared to baseline, but were not statistically significant. In each group there are significant differences between the time points (Friedman tests, *p* < 0.005 in each case). In group A, FMPS decreases after the first 6 months, and then remains relatively stable, while in groups B and C the decrease in the first 3 months is followed at 6 months by a return to values close to the initial ones (see Table 1 and Figure 6). The subjects presented with an average FMPS of 39.5 ± 20.36 in group A, 28.14 ± 24.59 in group B, and 28.43 ± 12.19 in group C, and an average FMBS of 66.62 ± 23.84 in group A, 51.59 ± 22.51 in group B, and 56.90 ± 19.60 in group C at the initial examination (Table 1). At the end of the 6-month period, the corresponding figures average FMPS of 24.00 ± 14.87 in group A, 22.82 ± 23.13 in group B, and 29.29 ± 14.20 in group C, and an average FMBS of 28.29 ± 10.87 in group A, 27.00 ± 21.60 in group B, and 28.57 ± 7.90 in group C, respectively.

The analysis of FMBS changes in test and control groups shows that there are no statistically significant differences between groups at any of the analyzed time points. The Friedman tests show in each case significant intragroup differences between the time points analyzed (*p* < 10^−6^ in each case). A substantial improvement was found in all groups at 3 months, followed by a slight increase at 6 months, which is marginally significant in group A (adjusted *p* = 0.051) and significant in groups B and C (see Figure 7). Significant improvements from baseline or differences between intragroup were observed in terms of FMBS and FMPS.

The proportion of sites with BOP is similar in the three groups at each analyzed time point. In each group, there is a significant decrease in the number of sites with BOP at 3 months, followed by a stabilization/possibly slight increase.

Likewise, regarding pocket closure (defined as the transition of sites with PPD > 5 mm or 4 mm with BOP to non-bleeding sites with PPD ≤ 4 mm), there were no statistically significant differences between groups both at baseline and at 3 and 6 months in all three groups; the proportion of sites without a therapeutic indication increases significantly at 3 months, then decreases slightly.

### 3.2. Results of Microbiological Tests

The microbiological results (*A. actinomycetemcomitans*, *P. gingivalis*, *P. intermedia*, *T. forsythia*, and *T. denticola*) were without statistical significance between all three groups at baseline and after 6 months (*p* = 0.190–0.859, respectively). The statistical analysis within the groups resulted in significantly lower detection scores for *P.g.*, *P.i.*, *T.f.*, and *T.d.* after six months for all groups. The *p*-values were between 0.007 and 0.029. There was no change in the bacterial counts of *A. actinomycetemcomitans* in the control group after six months. Relating to *A.a*., in the control group particularly, the numbers observed at baseline are comparable with those after 6 months, corresponding to Kruskal–Wallis tests for intergroup comparisons of the pathogen.

The detection scores in the intergroup analysis decreased; however, the differences between groups were not statistically significant.

## 4. Discussion

To our knowledge, the present study is the only clinical trial comparing the clinical and microbiological effects of Gelcide^®^ vs. a widely used SRD gel and a placebo for severe periodontitis treatment published so far. The present study identified similar improvements in clinical periodontal outcomes in 64 subjects treated with SI with or without a one-time administration of a locally delivered antibiotic at 6 months; as with other trials on LDD, the similarities of the results between test and control groups represent the main issue that has been identified in this research [16,44]. The purpose of the present study was to compare the effects of two locally delivered antibiotic gels (piperacillin plus tazobactam vs. doxycycline) at the end of SI in stage III–IV periodontitis patients. At any point during the trial, there were no adverse treatment effects identified. The goal was to evaluate the possible therapeutic benefits (in terms of reduction in PPD, reduction in BOP, and/or gains in CAL and a decrease in detection scores for the species *A.a.*, *P.g.*, *P.i.*, *T.f.*, *T.d.*).

The S3-level clinical guideline for the treatment of periodontitis [14] supports the use of adjuvant subgingival locally delivered antimicrobials following treatments in addition to SI at step 2 therapy (first nonsurgical phase). The infectious nature of periodontitis has led to the use of various antimicrobial substances to support mechanical debridement [45,46,47]. Since forty years ago, local antimicrobials have been used as a supplement to the treatment of periodontitis [30,48]. In local delivery systems, tetracycline hydrochloride, doxycycline, minocycline, metronidazole, and chlorhexidine have traditionally been the five main antibacterial medicines used [11]. Previous clinical studies evaluating the treatment of periodontitis with either “subgingival instrumentation” [14] alone [49,50,51] or with adjuncts, including locally administered antimicrobials such as minocycline [52,53,54], doxycycline [55,56], tetracycline [52,57], metronidazole [52], and chlorhexidine formulations [31,33], report considerable improvement of clinical parameters. According to the research on locally delivered antibiotic gels, they may be an essential component of periodontal therapy during both surgical and nonsurgical therapies [14,33,48]. In the present study, piperacillin plus tazobactam and doxycyclin gel were applied and compared with the placebo gel. The patients were followed for 6 months at 3-month intervals. The present treatment protocol included one round of application of piperacillin plus tazobactam, in periodontal pockets evaluated at 3–6 months after the application. Similar protocols of one local adjunctive application have been evaluated in the literature [16,44].

There are currently only two studies available assessing the clinical and microbiological effectiveness of locally applied piperacillin plus tazobactam in patients with periodontitis. They reported adjunctive benefits of modest magnitude and limited duration. Our results also showed improvements in clinical parameters after SI + locally delivered antibiotics; however, they were similar to SI alone.

This study proposed doxycycline and piperacillin plus tazobactam as effective antibacterial agents in subjects diagnosed with stage III-IV periodontitis with deep pockets, while in other similar studies, more shallow periodontal sites have been studied [58,59]. It is well known that the reduction in PPD in the range of 1.5 to 2.0 mm can be obtained by SI alone in deep periodontal pockets [60,61,62].

In the current study, the PPD decrease in all three study groups was similar to PPD reductions in earlier studies following combined SI and local antibiotics [63,64]. At 3 and 6 months, no statistically significant differences between 3 and 6 months were noted. The mean PPD at baseline was 5.89 ± 0.58 mm for the Gelcide^®^ group, 6.03 ± 0.58 mm for the Ligosan^®^ group, and 5.89 ± 0.63 mm for the placebo group, with a decrease at 6 months in Gelcide^®^ and Ligosan^®^ groups of Δ PPD = 0.2 mm (*p =* 0.837) when compared with the placebo group. These results indicate that SI with adjunctive delivery of Gelcide^®^ was equally effective as SI with Ligosan^®^ or placebo gel, which suggests that the adjunctive administration of antimicrobials had superior positive benefits in comparison to the placebo group, even if not statistically significant. This adjunctive effectiveness of the local application of an antibiotic (e.g., doxycycline) was also demonstrated by Eickholz et al. in 2002 in a similar study [65]. In comparison to SI with or without vehicle control, supplementary SRD (slow-released doxycycline) produced more beneficial PPD reductions gains in these individuals with untreated periodontitis, with an impact of 0.4 mm in SI alone, while SI plus adjunctive gel had an effect of 0.7 mm. However, in contrast to our study, this PPD reduction was statically significant. This difference can possibly be attributed to anatomical considerations, including the degree and topography of alveolar bone loss and attachment loss, which may limit the potential reduction in PPD. In our study, the selected teeth were not located in the anterior area (known as easier to maintain plaque-free when compared to distal areas); thus, the presence of plaque during the entire follow-up could not be entirely prevented.

Regarding BOP, the baseline proportion of sites was Gelcide^®^ 98.81%, Ligosan^®^ 100%, and placebo group 97.62%; the intragroup comparison showed a decrease after 3 months and after 6 months but with no statistical significance at the intergroup comparison. Similar to PPD, neither at the baseline nor during the final assessment after 6 months, were any statistically significant changes between the groups found for this parameter.

CAL showed significant differences in all groups at 3 and 6 months (*p* < 0.001 in each case). More precisely, there is a significant decrease in CAL between baseline and 3 months, followed by an increase at 6 months. Between-group comparisons revealed no statistically significant differences (*p* > 0.05) between baseline, 3, and 6 months regarding CAL. Other studies [4,19,31,33,66] showed that clinical parameters, such as BOP, PPD, and CAL gain improved in deep periodontal pockets (≥6 mm) after using locally delivered antimicrobials: chlorhexidine and doxycycline, respectively [63,64]; however, the results of the present study are in agreement with those of Lauenstein et al., 2013, that reported no statistically significant differences for PPD, BOP, and CAL between SI (scaling and root planing) alone and the local addition of piperacillin plus tazobactam, after 26 weeks of reevaluation [44].

In our study, patients’ levels of hygiene improved. The intragroup analysis showed a statistically significant reduction in FMPS at the 3-month time point (*p* = 0.015 and *p* = 0.004 at 6-months in the Gelcide^®^ group), but when intergroup comparisons were performed, differences did not show statistical significance. Interestingly, when compared with previous investigations of non-antibiotic antimicrobials adjunctive to SI, the PPD and BOP reductions in the present study are relatively similar [60,61,62,66,67,68,69].

The adjunctive use of piperacillin plus tazobactam in conjunction with SI in Step 3 therapy showed a tendency to produce better clinical and microbiological results, although not reaching a statistical difference. At 3 months after the application, the percentage of patients that achieved the endpoint of therapy (PPD ≤ 4 mm and BOP absence) was greater but not statistically significant in group A, and almost remained stable over time. Based on the evidence of the importance of this endpoint for the stability of the periodontal condition [14], it seems that piperacillin plus tazobactam adjunctive to SI during Step 3 therapy increases the possibility of obtaining stability, although not reaching statistical importance. As expected, the greatest improvement was noticed at 3 months after the application, both in our study and in previous studies [16,44].

Regarding pocket closure in the present study, a reduction of 65.48% was recorded after 3 months in group A, 60.23% in group B, and 58.33% in the placebo group, respectively. At the end of the follow-up, the pocket closure attained 46.43% in group A, 43.18% in group B, and 36.90% in the placebo group, respectively. The reduction was almost comparable with the reduction after nonsurgical periodontal therapy reported by Tomasi et al. (expressed as 50% of the tooth sites with an initial PD ≥ 5 mm after 3 months), in a study using locally delivered doxycycline, as well [19]. Overall, neither treatment resulted in the complete resolution of all residual pockets. This could be linked to the finding that the presence of plaque, even on a single tooth, had a negative impact on the clinical outcome in nonsurgical periodontal treatments of periodontitis [70].

In our trial, the reduction in biofilm was achieved by using repeated USI at sites with PPD ≥4 mm at the 3- and 6-month follow-up for both the test and control groups, while the gel application was performed only once, counting on the duration of the adjunctive advantage of Gelcide^®^ to SI, due to its persistence on the root surfaces.

Due to the broad range of pertinent factors that vary from research to research, such as the included teeth, treatment duration, and study population, it is rather challenging to evaluate the clinical results obtained by various controlled clinical trials involving topical administration adjuvant to SI. In general, the outcomes for SI alone, when compared to SI combined with LDD at 3 and 6 months after therapy, were less favorable in all studies on adjunctive topical delivery of antibiotics evaluated [36,52,53,65,71], on par with those observed in our investigation. The benefit of using antimicrobial gels subgingivally in addition to SI failed to exceed *p* values < 0.05 at the study’s endpoint [36,52,71]. However, these agents were deemed to be useful complements to SI [65].

There are currently few clinical studies examining the effectiveness of treating stage III-IV periodontitis patients with a single dose of piperacillin plus tazobactam to reduce bacterial counts in periodontal pockets [64,72]. In our trial, *P.g.*, *T.d.*, *T.i.*, *A.a.*, and *P.i*. were chosen as the target pathogens. The intragroup comparison showed significantly lower detection scores for *P.g.*, *P.i.*, *T.f.*, and *T.d.* between the baseline and the 6-month time point. More precisely, at 6 months after treatment, the intragroup analysis revealed a statistically significant decrease in detection scores from baselines for *P.i.* (group A and B with *p* values of 0.020 and 0.007), *P.g.* (group A, B, and C with *p* values of 0.02, 0.006, and 0.026, respectively), *T.f.* (group A, B, and C with *p* values of 0.014, 0.001, and 0.006, respectively), and *T.d.* (group A and B with *p* values of 0.029 and 0.003). Similar results were obtained in another study from 2013, regarding the reduction in pathogens (including *P.g.*, *T.f.*, or *A.a.*) after 26 weeks (~6 months) following local application of piperacillin plus tazobactam, in patients with chronic periodontitis [44]. Relating to *A.a.*, in the control group particularly, the numbers observed at baseline, are comparable with those after 6 months, indicating that SI alone is unlikely to dramatically diminish or eradicate this pathogen. This finding is consistent with numerous research studies that claim *A.a.* is challenging to treat with just mechanical debridement [72,73,74]. At the intergroup comparison, the differences in detection scores of the target pathogen were not statistically significant, on par with previous studies [16,44].

The inclusion of the antibiotics in an insoluble biodegradable pellicle generated by mixing with a volatile polymer could be one of the possible benefits of the piperacillin plus tazobactam formulation. The manufacturers claim that this novel presentation is able to keep the antibiotic active for 7–12 days after the application in the periodontal pocket; however, these claims are not backed by published research so far. Further studies should clarify this aspect, as well.

The major strength of this study is the fact that this is the first study comparing piperacillin plus tazobactam gel with doxycycline and a placebo gel as an adjunct to subgingival instrumentation. The limitations of this study include the unknown duration of the effect of the adjunctive topical administration of antibiotics and the relatively short follow-up period. Despite the effect’s decrease over time, the improvements noted during the follow-up period may be sufficient to demonstrate some benefit of the treatment, as LDDs are anticipated to have the strongest antimicrobial effects during the first few days following treatment, when still detectable in subgingival sites [70,75]. This may indicate that clinical benefits of Gelcide^®^ are comparable with those of widely used local doxycycline periodontal products. This finding is important when taking into consideration that the cost per single application of Gelcide^®^ is approximately equal to that of Ligosan^®^, if the manufacturer’s instructions are respected. The single application of the drug, dictated in our study by economic considerations, could also be seen as a limitation of this study. Potential indirect biases of this study could be considered that systemic diseases and medication should have been considered, and tooth types for each patient were not specified.

## 5. Conclusions

The groups had similar results, and subgingival instrumentation can be executed without adjunctive antimicrobials, reducing the costs for the patient and the working time/load of the professional.

## Figures and Tables

**Figure 1 medicina-59-00303-f001:**
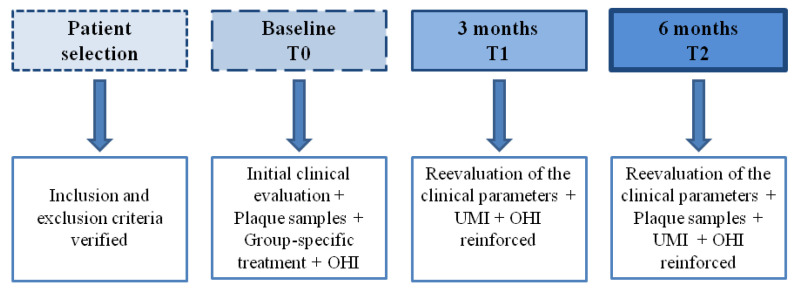
Treatment timeline.

**Figure 2 medicina-59-00303-f002:**
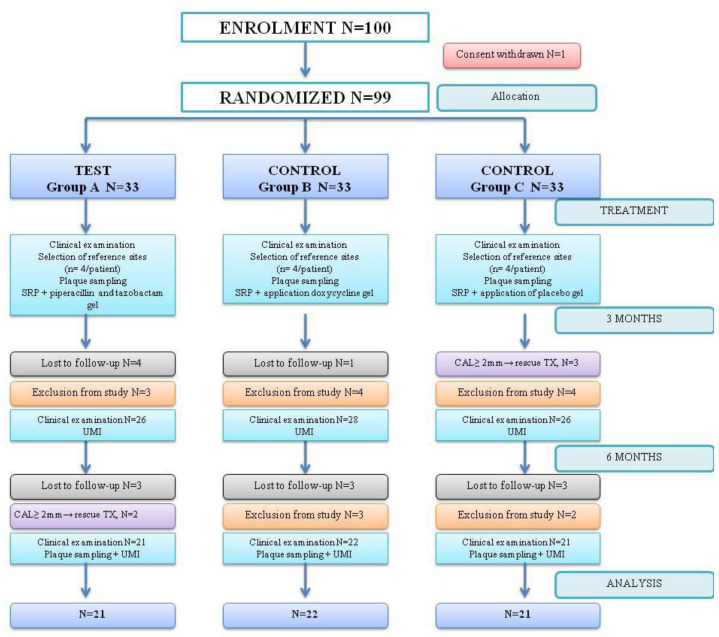
CONSORT Diagram and patient allocation.

**Figure 3 medicina-59-00303-f003:**
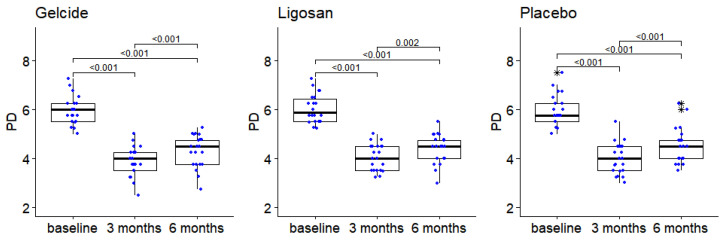
The evolution of the PPD between successive time points in the three groups (numbers in brackets represent *p*-values of Wilcoxon tests for pairwise intragroup comparisons). Boxplot depicts median and interquartile range, large stars represent outliers; data points displayed as superimposed dot plots.

**Figure 4 medicina-59-00303-f004:**
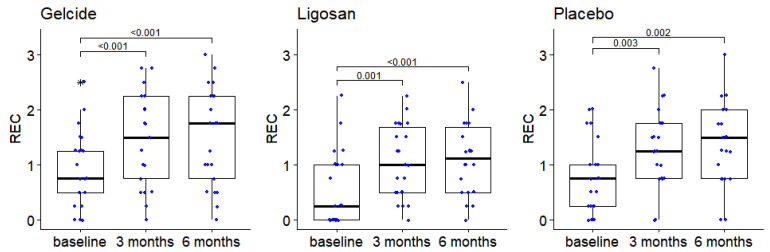
The evolution of the REC between successive time points in the three groups (numbers in brackets represent *p*-values of Wilcoxon tests for pairwise intragroup comparisons; *p*-values > 0.05 are not displayed), boxplot depicts median and interquartile range, large stars represent outliers; data points displayed as superimposed dot plots.

**Figure 5 medicina-59-00303-f005:**
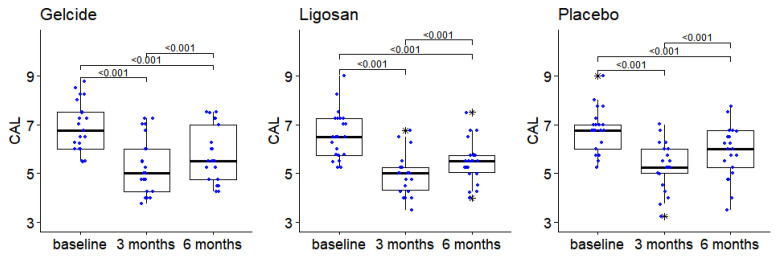
The evolution of the CAL between successive time points in the three groups (numbers in brackets represent *p*-values of Wilcoxon tests for pairwise intragroup comparisons; *p*-values > 0.05 are not displayed), boxplot depicts median and interquartile range, large stars represent outliers; data points displayed as superimposed dot plots.

**Figure 6 medicina-59-00303-f006:**
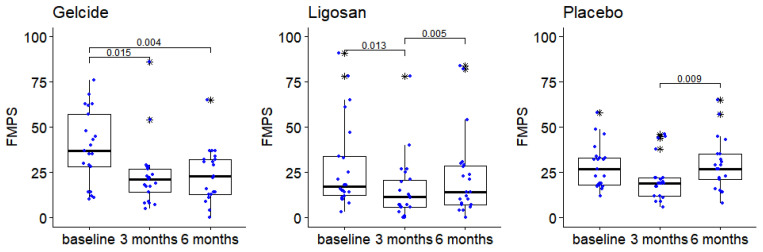
The evolution of the FMPS between successive time points in the three groups (numbers in brackets represent *p*-values of Wilcoxon tests for pairwise intragroup comparisons; *p*-values > 0.05 are not displayed), boxplot depicts median and interquartile range, large stars represent outliers; data points displayed as superimposed dot plots.

**Figure 7 medicina-59-00303-f007:**
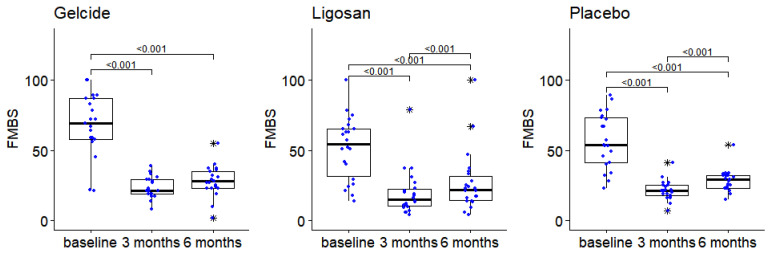
The evolution of the FMBS between successive time points in the three groups (numbers in brackets represent *p*-values of Wilcoxon tests for pairwise intragroup comparisons; *p*-values > 0.05 not displayed), boxplot depicts median and interquartile range, large stars represent outliers; data points displayed as superimposed dot plots.

**Table 1 medicina-59-00303-t001:** Demographic data and full-mouth clinical parameters at baseline and follow-up visits in the three groups.

Variable	Time Point	Group A (*n* = 21)	Group B (*n* = 22)	Group C (*n* = 21)	*p*-Values
*Age* (*years*, *mean* ± *sd*)	Baseline	50.71 *±* 9.56	47.32 ± 8.08	49.95 ± 6.61	0.489 ^a^
*Sex = female* (*n*, %)	Baseline	8 (38.10%)	13 (59.09%)	14 (66.67%)	0.156 ^b^
*Smokers* (*n*, %)	Baseline	9 (42.86%)	8 (36.36%)	9 (42.86%)	0.881^b^
PPD (*n*, %) 5 mm 6 mm 7 mm 8 mm	Baseline Baseline Baseline Baseline	28 (33.33%) 34 (40.48%) 18 (21.43%) 4 (4.76%)	32 (36.36%) 29 (32.96%) 19 (21.59%) 8 (9.09%)	28 (33.33%) 35 (41.67%) 16 (19.05%) 5 (5.95%)	0.853 ^a^
PPD (mm *± sd*)	Baseline 3 months Difference to baseline 6 months Difference to baseline	5.98 ± 0.583.88 ± 0.632.10 ± 0.444.30 ± 0.671.68 ± 0.49	6.03 ± 0.584.03 ± 0.542.00 ± 0.454.40 ± 0.591.67 ± 0.47	5.98 ± 0.634.00 ± 0.641.98 ± 0.594.51 ± 0.751.47 ± 0.62	0.936 ^a^ 0.766 ^a^ 0.919 ^a^ 0.837 ^a^ 0.410 ^a^
REC (mm *± sd*)	Baseline 3 months Difference to baseline 6 months Difference to baseline	0.93 ± 0.691.45 ± 0.850.52 ± 0.411.50 ± 0.900.57 ± 0.43	0.55 ± 0.671.05 ± 0.660.50 ± 0.471.09 ± 0.670.55 ± 0.46	0.77 ± 0.681.32 ± 0.700.55 ± 0.611.39 ± 0.750.62 ± 0.61	0.119 ^a^ 0.216 ^a^ 0.926 ^a^ 0.231 ^a^ 0.955 ^a^
CAL (mm *± sd*)	Baseline 3 months Difference to baseline 6 months Difference to baseline	6.90 ± 1.035.33 ± 1.171.57 ± 0.515.80 ± 1.141.10 ± 0.54	6.58 ± 0.985.03 ± 0.901.55 ± 0.575.49 ± 0.861.09 ± 0.60	6.75 ± 0.925.32 ± 0.971.43 ± 0.575.90 ± 1.080.85 ± 0.61	0.575 ^a^ 0.475 ^a^ 0.573 ^a^ 0.315 ^a^ 0.203 ^a^
FMPS (*± sd*)	Baseline 3 months Difference to baseline 6 months Difference to baseline	39.05 ± 20.3623.14 ± 17.9715.91 ± 20.9224.00 ± 14.8715.05 ± 19.27	28.14 ± 24.5915.77 ± 17.3012.37 ± 20.5322.82 ± 23.135.32 ± 24.93	28.43 ± 12.1921.38 ± 11.927.05 ± 17.9329.29 ± 14.20−0.86 ± 20.86	0.074 ^a^ 0.045 ^a^ 0.396 ^a^ 0.072 ^a^ 0.093 ^a^
FMBS (*± sd*)	Baseline 3 months Difference to baseline 6 months Difference to baseline	66.62 ± 23.8423.33 ± 7.5343.29 ± 22.4128.29 ± 10.8738.33 ± 21.61	51.59 ± 22.5119.59 ± 16.3432.00 ± 23.8727.00 ± 21.6024.59 ± 26.89	56.90 ± 19.6021.48 ± 16.9735.42 ± 18.9728.57 ± 7.9028.33 ± 19.95	0.077 ^a^ 0.057 ^a^ 0.188 ^a^ 0.152 ^a^ 0.113 ^a^
BOP	Baseline 3 months 6 months	83/84 (98.81%) 25/84 (29.76%) 27/84 (32.14%)	88/88 (100%) 26/88 (29.55%) 30/88 (34.09%)	82/84 (97.62%) 29/84 (34.52%) 35/84 (41.67%)	- 0.732 ^b^ 0.396 ^b^
Pocket closure (%)	Baseline 3 months 6 months	0/84 55/84 (65.48%) 39/84 (46.43%)	0/88 53/88 (60.23%) 38/88 (43.18%)	0/84 49/84 (58.33%) 31/84 (36.90%)	- 0.615 ^b^ 0.446 ^b^

^a^ Kruskal–Wallis test; ^b^ chi-square test.

## Data Availability

Not applicable.

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
