# Peer review of "A Placebo-Controlled Trial to Evaluate Two Locally Delivered Antibiotic Gels (Piperacillin Plus Tazobactam vs. Doxycycline) in Stage III–IV Periodontitis Patients"

_medicina, 2023, doi:10.3390/medicina59020303_

Round 1

Reviewer 1 Report

The manuscript details a  well designed clinical study that seeks to determine if a local delivery system containing piperacillin and tazobactam is effective in managing Stage III-IV periodontitis.  It compares this local delivery system to a doxycycline based local delivery system and a placebo gel control.  The strengths of the paper are its design (particularly use of placebo gel control), the carefully outlined methods section and concise conclusions. 

1. My major concern with the manuscript is the presentation of the results.  For example, Table 2, which outlines the microbial results is unnecessarily detailed and lends little to the understanding of the data.  These data can be easily presented in the text by focusing on those species that changed in a statistically significant way at 6 months.    These microbial  data should  be more succinctly presented to improve the manuscript considerably  (as they are  presented in the discussion section Page 15, 528-545). 

2,  A second concern with the manuscript is the focus (in both data section and discussion) on the intragroup changes at 6 months.  These changes are well known in the literature.  Changing the results to focus predominantly on the admittedly negligible intergroup differences (which is the stated aim of the manuscript)  would add brevity and clarity to manuscript.  

Author Response

Reviewer’s concern #1:

  1. My major concern with the manuscript is the presentation of the results.  For example, Table 2, which outlines the microbial results is unnecessarily detailed and lends little to the understanding of the data.  These data can be easily presented in the text by focusing on those species that changed in a statistically significant way at 6 months. These microbial  data should  be more succinctly presented to improve the manuscript considerably  (as they are  presented in the discussion section Page 15, 528-545). 

Our response: First of all, thank you for the kind comments regarding our study. Secondly, we thank you for your suggestions which improve our manuscript substantially. We eliminated Table 2, as suggested, and the data are presented in the text below.

Revised text:

3.2. Results of microbiological tests

The microbiological results (A. actinomycetemcomitans, P. gingivalis, P. intermedia, T. forsythia, T. denticola) were without statistical significance between all three groups at baseline and after 6 months (p = 0.190 – 0.859, respectively). The statistical analysis within the groups resulted in significantly lower detection scores for P.g., P.i., T.f., and T.d. after six months for all groups. The p - values were between 0.007 and 0.029. There was no change of the bacterial counts of A. actinomycetemcomitans in the control group after six months. Relating to A.a., in the control group particularly, the numbers observed at baseline, are comparable with those after 6 months. Corresponding to Kruskal-Wallis tests for intergroup comparisons of pathogen.

Reviewer’s concern #2:

2, A second concern with the manuscript is the focus (in both data section and discussion) on the intragroup changes at 6 months.  These changes are well known in the literature.  Changing the results to focus predominantly on the admittedly negligible intergroup differences (which is the stated aim of the manuscript)  would add brevity and clarity to manuscript.  

Our response: Thank you for your kind remarks. We edited the data section and discussion accordingly.

Revised text: We edited the data section and the Discussion section according to your request, starting with line 326 – see into the revised manuscript.

Reviewer 2 Report

Manuscript: medicina-2142403

Title: A placebo-controlled trial to evaluate two locally delivered antibiotic gels (piperacillin plus tazobactam vs. doxycycline) in stage III–IV periodontitis patients.

This present manuscript evaluated the efficacy of the combination of piperacillin plus tazobactam vs. doxycycline compared to a placebo gel adjunctive to subgingival instrumentation in stage III–IV periodontitis.

This reviewer understands all the authors' efforts in carrying out the research and preparing the manuscript; however, some considerations are necessary.

Comments to Authors

Abstract

Page 1, Lines 39 and 40

Conclusion:

…within its limits it failed to indicate inter-group statistically significant differences between the groups in subjects with severe periodontitis…

I recommend just reporting, "the groups had similar results..."

Materials and methods:

Study design

I suggest that all information related to the calculation of the sample size be included at the beginning of the materials and methods.

Page 3, line 136

“…Prior to the study, the examiners were trained to complete…”

The authors should describe in more detail how the training process was carried out for the evaluation of the clinical parameters.

Did the authors perform calibration? The calibration process involves applying statistical tests to confirm that the reproducibility of the data is consistent.

For some periodontal parameters, calibration is not possible, only training.

Therefore, a more detailed description is necessary.

Page 3, lines 142 to 145

“…All subjects were diagnosed with periodontitis stage III and IV. All participants fulfilled the following inclusion criteria: subjects aged over 25 years, at least 8 sites with PD ≥ 5 mm and showing bleeding on probing, clinical attachment loss ≥3 mm, patients who have not undergone periodontal therapy in the last 12 months.”

I recommend that the authors check the criteria used for the classification of periodontitis stages. Not following reference 21.

21. Tonetti MS, Greenwell H, Kornman KS. Staging and grading of periodontitis: Framework and proposal of a new classification and case definition. J Periodontol. 2018 Jun;89 Suppl 1:S159-S172. doi: 10.1002/JPER.18-0006

Stage III: Severe periodontitis with potential for additional tooth loss.

-Interdental CAL at site of greatest loss: ≥5mm

-Radiographic bone loss: Extending to mid-third of root and beyond

-Tooth loss: Tooth loss due to periodontitis of ≤4 teeth

-Probing depth ≥6 mm

-Vertical bone loss ≥3 mm

-Furcation involvement Class II or III

-Moderate ridge defect

Stage IV: Advanced periodontitis with extensive tooth loss and potential for loss of dentition

-Interdental CAL at site of greatest loss: ≥5mm

-Radiographic bone loss: Extending to mid-third of root and beyond

-Tooth loss: Tooth loss due to periodontitis of ≥5 teeth

-Probing depth ≥6 mm

-Vertical bone loss ≥3 mm

-Furcation involvement Class II or III

-Moderate ridge defect

- Need for complex rehabilitation due to:

Masticatory dysfunction Secondary occlusal trauma (tooth

mobility degree ≥2)

Severe ridge defect

Bite collapse, drifting, flaring Less than 20 remaining teeth

(10 opposing pairs)

Page 4, lines 157 and158

“…Probing Pocket Depth (PPD) evaluation on the vestibular and oral surfaces was performed halfway between the line angles”

I don't understand this sentence. Did the authors not perform a periodontal examination on the lingual surfaces?

Please clarify.

Page 4, lines 160 and 161

“…immediate vicinity of the contact point with moderate pressure”

How did the authors define "moderate pressure"? How was the probing pressure controlled, was any training done (eg using a digital scale or pressure-controlled periodontal probe)?

Page 4, lines 170 and 171

“…The patients were excluded from the study in case of progressive attachment loss of 2 mm or more between two subsequent evaluation timepoints.

The authors should explain the reasons for excluding patients with clinical attachment loss of 2mm or more.

I think that excluding these patients could create biases in the study. But the authors can justify presenting the number of patients excluded from each group for this reason

Page 4, lines 190 and 191

“…commercial kit (micro-IDent A 190 Test).”

Please include a complete description: manufacturer, city, country, and if possible, website.

Page 4, lines 204 and 205

“…The differences between the original packages of the two gels precluded 204 the blinding of the operator…”

Could the fact that the operator was not "blinded" generate some bias in the study? This point should be addressed in the limitations of the study in the discussion section.

Authors should consider including this information in the manuscript title "a single-blinded study".

Page 5, line 208

“…ultrasonic SI…”

Authors should provide more details about the ultrasonic device used: magnetostrictive or piezoelectric; power; tip type, and manufacturer.

Page 5, line 215

“…including oral hygiene instruction…”

The authors should provide more details about the strategies used for oral hygiene orientation: one-time, brushing technique, dentifrice, and direct and indirect methods.

Page 5, line 220

“…followed by manual SI with Gracey curettes; #5/6; 7/8; 11/12; 13/14…”

I recommend that the authors include the criteria used to determine whether the subgingival instrumentation procedure was adequate. It was either the operator himself who checked whether the subgingival deposits had been removed or a second operator did the checking. 

Please include this information.

Page 5, lines 233 and 234

“…Group C (negative control), a placebo gel, with similar aspect and consistency as the products used in Groups A & B…”

Here two points must be considered.

First, the authors should check the definitions of the negative control group and placebo control group; they are different and it seems to me the authors use the terms as synonyms:

-       Negative control group: In this type of control group, the participants are not given a treatment. The experimental group can then be compared to the group that did not experience any change or results.

-       Placebo control group: This type of control group receives a placebo treatment that they believe will have an effect. This control group allows researchers to examine the impact of the placebo effect and how the experimental treatment compared to the placebo treatment.

Second, in lines 201 and 202 “…Due to the particular physical nature and package of the products, only the patients and the evaluator were blinded…”

Therefore, if the two active gels were different, shouldn't the present study have two placebos? With one placebo simulating each of the different gels.

According to the description of the materials and methods this care was not taken. Therefore, the authors should justify this point, since it may generate a "nocebo effect" inserting biases in the study.

Statistical Analysis

Page 5, line 248

“…are needed to detect a significant mean difference of 1 mm in…”

What were the criteria used by the authors to define that 1 mm would be the minimum difference to be considered clinically relevant in reducing clinical pocket depth? A 1mm reduction in clinical pocket depth can be achieved with supragingival instrumentation and hygiene orientation alone. 

Please clarify this point.

Page 6, lines 256 to 259

“…using Kruskal–Wallis tests, with post–hoc Mann–Whitney pairwise tests, as necessary. Proportions were compared by chi square tests. Assessment of intragroup differences between successive time points for quantitative variables was performed using Friedman tests, with subsequent Wilcoxon signed-rank tests for pairwise…”

The authors conducted the data analysis only with non-parametric models (Kruskal-Wallis, Mann-Whitney, Friedman and Wilcoxon); however, according to Cohen (2001):

“For the analysis of dental indices, even those that are not normally distributed, parametric methods are generally valid, usually more powerful, and always more versatile than non-parametric alternatives. Routine rejection of parametric tests because data are non-normal ignores the documented robustness of these methods under common design features of dental research. Rejection of parametric tests because data are ordinal is not required by statistical theory. Selection of appropriate statistical methods should not be dictated by measurement-scale typology but should be guided by careful evaluation of the scientific meaning and the empirical value of those measurements.”

Reference

Cohen ME. Analysis of ordinal dental data: evaluation of conflicting recommendations. J Dent Res. 2001 Jan;80(1):309-13. doi: 10.1177/00220345010800010301. PMID: 11269721.

Therefore, as a suggestion (if indeed) I recommend that the authors think of alternatives for analyzing the data so that the results can be confirmed.

Results

Page 6, lines 271 and 272

“No side effects or negative impacts connected to the treatment methods were found during the study period.”

I recommend that the authors include more detail in this paragraph. The authors should consider reporting pain after subgingival instrumentation and dentinal hypersensitivity.

Page 8, Table 1

In the probing pocket depth (PPD) the authors need to include the result of the statistical test to confirm that the distribution of the probing pocket depth (PPD) in millimeters confirms the percentage distribution.

Page 8, Figure 3

The authors need to make it clear in the legend of figure 3 what the graph represents: mean and standard deviation or median and interquartile range. I recommend that all dots be included in the figure, not just the outliers.

Please check, Drummond & Vowler (2011): “The present article encourages authors to present data clearly, preferably as a dot plot, so that the distribution of the values can be recognized”.

The p-values presented have 6 decimal places, I recommend a maximum of 4 decimal places.

Change the information about the groups: instead of using Group A, I recommend changing the treatment used.

Reference:

Drummond GB, Vowler SL. Show the data, don't conceal them. Br J Pharmacol. 2011 May;163(2):208-10. doi: 10.1111/j.1476-5381.2011.01251.x. PMID: 21501140.

Page 8, lines 309 and 310

“However, despite 309 these modifications of the parameters, the variations among the groups never reached statistical significance…”

This reviewer has a slightly different opinion than the authors regarding the statement. The impression I get is that the authors are justifying the fact that the results did not reach significant differences ("the sacred p-value"). I simply state the following "the groups had similar results".

Page 9, Figure 4

Check the same considerations made for figure 3.

Page 10, Figure 5

Check the same considerations made for figure 3.

Page 10, line 304

“…the values in group B being apparently…”

This statement is not adequate, since the results had no significant differences. We cannot state that apparently, the results are lower. Since the test had no significant difference, we can only state that they were similar.

Page 10, Figure 6

Check the same considerations made for figure 3.

Page 11, Figure 7

Check the same considerations made for figure 3.

Page 11, Table 2

Please explain the missing values in Table 2. Information should be added at the bottom of the table.

Discussion

Page 13, lines 434 and 435

“The present treatment protocol included one round of application of piperacillin plus tazobactam, in periodontal pockets evaluated at 3-6 months the application.…”

The authors applied the gels in a single round. This point should be further explored in the discussion. If the study had performed a second instrumentation session with antimicrobial gel application at 3 months could the results be different?

Page 13, lines 434 and 435

The authors did not adequately explain both the strengths and weaknesses of the observations should be discussed."

Questions regarding the costs (dollar or euro) of antimicrobial gels for use in periodontal therapy should be discussed.

The perceptions of the patient (patient-reported outcomes and patient-reported experience) and the operators should also be presented and discussed.

Check:

McGuire MK, Scheyer ET, Gwaltney C. Commentary: incorporating patient-reported outcomes in periodontal clinical trials. J Periodontol. 2014 Oct;85(10):1313-9. doi: 10.1902/jop.2014.130693.

I recommend that the authors use NNT (number needed to treat) calculations to enhance determining the clinical relevance of this periodontal research finding. The authors could consider it as a secondary analysis of the data so that it can be discussed. I believe that the use of NNT can contribute to the clinical understanding of the results.

Check:

Greenstein G, Nunn ME. A method to enhance determining the clinical relevance of periodontal research data: number needed to treat (NNT). J Periodontol. 2004 Apr;75(4):620-4. doi: 10.1902/jop.2004.75.4.620.

The biases of the study are not discussed.

Conclusion:

The conclusion does not adequately reflect what the results showed.

The authors should elaborate a conclusion that better reflects the results, i.e., the groups had similar results, and according to the results we can use the subgingival instrumentation methods without the need to use antimicrobials, reducing the costs for the patient and the working time of the professional.

Decision:

The manuscript is well written and the research well designed, but several points should be considered by the authors. Therefore, I recommend that the manuscript be sent to the authors to answer the questions and make the suggested changes.

After all, I can recommend the manuscript for the authors' review, only after the answer or justification can I make a final decision.

Author Response

Reviewer’s concern #1:  Abstract

Page 1, Lines 39 and 40

Conclusion:

…within its limits it failed to indicate inter-group statistically significant differences between the groups in subjects with severe periodontitis…

I recommend just reporting, "the groups had similar results..."

Our response: First of all, thank you for the kind comments and suggestions to improve our paper. We edited the text as suggested.

Revised text:

Conclusion: The groups had similar results, and subgingival instrumentation can be executed  without adjunctive antimicrobials, reducing the costs for the patient and the working time/load of the professional.

Reviewer’s concern #2:

Materials and methods:

Study design

I suggest that all information related to the calculation of the sample size be included at the beginning of the materials and methods.

Our response: Thank you for the kind suggestion; we moved the sample size calculation to the beginning of M&M.

Revised text: “…allocation ratio.

It was established that 19 patients per group are needed to detect a significant mean difference of 1 mm in PPD reduction between groups, assuming a common standard deviation of 1 mm, 80% power, and a significance level of 0.05. The Pitman asymptotic relative efficiency correction was used in the sample size calculations to account for the use of nonparametric tests. Considering an anticipated drop-out rate of ~10%, it was decided to enroll at least 21 patients in each group.

The study included ....”

Reviewer’s concern #3:

Page 3, line 136

“…Prior to the study, the examiners were trained to complete…”

The authors should describe in more detail how the training process was carried out for the evaluation of the clinical parameters.

Did the authors perform calibration? The calibration process involves applying statistical tests to confirm that the reproducibility of the data is consistent.

For some periodontal parameters, calibration is not possible, only training.

Therefore, a more detailed description is necessary.

Our response: Thank you for your question. A calibration of the blinded examiner has been performed before. The term “trained” has been deleted. The intra-examiner calibration for reliability testing resulted in κ=0.92 for repeated measurements of PPD and CAL in two quadrants of five patients, other than the patients recruited for the study.

Revised text: “Prior to the study, the examiner (specialist of Periodontology) was calibrated, the intra-examiner calibration for reliability testing resulted in κ=0.92 for repeated measurements of PPD and CAL in two quadrants of five patients, other than the patients recruited for the study (to complete the evaluations needed for this study in a reliable and accurate manner that is consistent with current standards for clinical periodontal studies).”

Reviewer’s concern #4: Page 3, lines 142 to 145

“…All subjects were diagnosed with periodontitis stage III and IV. All participants fulfilled the following inclusion criteria: subjects aged over 25 years, at least 8 sites with PD ≥ 5 mm and showing bleeding on probing, clinical attachment loss ≥3 mm, patients who have not undergone periodontal therapy in the last 12 months.”

I recommend that the authors check the criteria used for the classification of periodontitis stages. Not following reference 21.

  1. Tonetti MS, Greenwell H, Kornman KS. Staging and grading of periodontitis: Framework and proposal of a new classification and case definition. J Periodontol. 2018 Jun;89 Suppl 1:S159-S172. doi: 10.1002/JPER.18-0006

Stage III: Severe periodontitis with potential for additional tooth loss.

-Interdental CAL at site of greatest loss: ≥5mm

-Radiographic bone loss: Extending to mid-third of root and beyond

-Tooth loss: Tooth loss due to periodontitis of ≤4 teeth

-Probing depth ≥6 mm

-Vertical bone loss ≥3 mm

-Furcation involvement Class II or III

-Moderate ridge defect

Stage IV: Advanced periodontitis with extensive tooth loss and potential for loss of dentition

-Interdental CAL at site of greatest loss: ≥5mm

-Radiographic bone loss: Extending to mid-third of root and beyond

-Tooth loss: Tooth loss due to periodontitis of ≥5 teeth

-Probing depth ≥6 mm

-Vertical bone loss ≥3 mm

-Furcation involvement Class II or III

-Moderate ridge defect

- Need for complex rehabilitation due to:

Masticatory dysfunction Secondary occlusal trauma (tooth

mobility degree ≥2)

Severe ridge defect

Bite collapse, drifting, flaring Less than 20 remaining teeth

(10 opposing pairs)

Our response:  Thank you for your remarks. We apologize for a typing error; the clinical attachment loss is ≥ 5 mm, not ≥ 3 mm.

Revised text: “…All subjects were diagnosed with periodontitis stage III and IV. All participants fulfilled the following inclusion criteria: subjects aged over 25 years, at least 8 sites with PD ≥ 5 mm and showing bleeding on probing, clinical attachment loss ≥ 5 mm, patients who have not undergone periodontal therapy within the last 12 months.”

Reviewer’s concern #5: Page 4, lines 157 and158

“…Probing Pocket Depth (PPD) evaluation on the vestibular and oral surfaces was performed halfway between the line angles”

I don't understand this sentence. Did the authors not perform a periodontal examination on the lingual surfaces?

Please clarify.

Our response: In our understanding, the word “oral” is used both for “palatal” and “lingual” aspects of the teeth in the maxilla or mandibular, respectively. We replaced “oral” with “palatal” and “lingual” in the text.

Revised text: “Probing Pocket Depth (PPD) evaluation on the vestibular, palatal and lingual surfaces was performed halfway between the line angles”

Reviewer’s concern #6: Page 4, lines 160 and 161

“…immediate vicinity of the contact point with moderate pressure”

How did the authors define "moderate pressure"? How was the probing pressure controlled, was any training done (eg using a digital scale or pressure-controlled periodontal probe)?

Our response: By “moderate pressure” we understand the standard probing pressure of 20-25N, as recommended by the British Society of Periodontology in 2001 and by: Polson et al., 1980; Garnick et al., 1989; Armitage et al., 1977. As the examiner was a specialist of periodontology, we assume his considerable experience allowed him to exert an adequate pressure when performing the probing. The word “moderate” is employed as such in numerous similar studies to designate the adequate probing pressure.

A force of 20-25N cause minimal discomfort and still enables accurate diagnostic readings according to Polson et al., 1980; Garnick et al., 1989; Armitage et al., 1977”.

* Al Shayeb KN, Turner W, Gillam DG. Accuracy and reproducibility of probe forces during simulated periodontal pocket depth measurements. Saudi Dent J. 2014 Apr;26(2):50-5. doi: 10.1016/j.sdentj.2014.02.001. Epub 2014 Mar 18. PMID: 25408596; PMCID: PMC4229682.

Revised text: “…immediate vicinity of the contact point with optimal pressure”

Reviewer’s concern #7: Page 4, lines 170 and 171

“…The patients were excluded from the study in case of progressive attachment loss of 2 mm or more between two subsequent evaluation timepoints.

The authors should explain the reasons for excluding patients with clinical attachment loss of 2mm or more.

I think that excluding these patients could create biases in the study. But the authors can justify presenting the number of patients excluded from each group for this reason

Our response: There were three reasons for this choice: first - the primary goal of the therapy was to prevent the disease progression; the second reason was to reduce the risk of tooth loss as a result of the disease, and finally, to prevent its recurrence. So, if during the 6-month follow-up, an attachment loss > 2 mm was observed, the patients immediately received re-instrumentation of the affected sites, and this could have been considered a bias indeed. We followed your advice and presented the number of patients excluded from each group with regard to this criterion in the study flowchart in Figure 2.

Revised text: “…The patients were excluded from the study in case of progressive attachment loss of 2 mm or more between two subsequent evaluation timepoints, as re-instrumentation of the affected sites was deemed necessary.”

Reviewer’s concern #8: Page 4, lines 190 and 191

“…commercial kit (micro-IDent A 190 Test).”

Please include a complete description: manufacturer, city, country, and if possible, website.

Our response: Thank you for careful reading, here is the complete description: micro-IDent®, produced by Hain Lifescience, Hardwiesenstraße 1, 72147 Nehren, Germany, website: https://www.hain-lifescience.de/en/company/company-profile.html

Revised text: micro-IDent® (Hain Lifescience, Nehren, Germany)

Reviewer’s concern #9: Page 4, lines 204 and 205

“…The differences between the original packages of the two gels precluded the blinding of the operator…”

Could the fact that the operator was not "blinded" generate some bias in the study? This point should be addressed in the limitations of the study in the discussion section.

Authors should consider including this information in the manuscript title "a single-blinded study".

Our response: Indeed, this might be considered as a limitation, due to the differences in the design of the packages and to the obvious differences in consistency and color between the products. Thus, the operator could identify very easy the product he applied. However, the patient, the examiner, and the data collector were blinded. We modified the text accordingly.

Revised text: “…The differences between the original packages of the gels and the obvious differences in consistency and color precluded the blinding of the operator…”

Reviewer’s concern #10: Page 5, line 208

“…ultrasonic SI…”

Authors should provide more details about the ultrasonic device used: magnetostrictive or piezoelectric; power; tip type, and manufacturer.

Our response: Thank you. The ultrasonic device used was: EMS Piezon® Master, EMS, Nyon, Switzerland, using fine subgingival inserts PS (Perio Slim) EMS, Nyon, Switzerland.

Revised text: “…ultrasonic SI using EMS Piezon® Master and PerioSlim inserts (EMS, Nyon, Switzerland)”

Reviewer’s concern #11: Page 5, line 215

“…including oral hygiene instruction…”

The authors should provide more details about the strategies used for oral hygiene orientation: one-time, brushing technique, dentifrice, and direct and indirect methods.

Our response: More details were added in this paragraph, as suggested.

Revised text: “Standard oral-health instructions were recommended: tooth brushing, either with manual or powered toothbrushes, minimum 2 minutes twice per day, interdental cleaning with interdental brushes. Instructions were personalized according to the patient’s need to obtain the best plaque control. No antiseptics were recommended.”

Reviewer’s concern #12: Page 5, line 220

“…followed by manual SI with Gracey curettes; #5/6; 7/8; 11/12; 13/14…”

I recommend that the authors include the criteria used to determine whether the subgingival instrumentation procedure was adequate. It was either the operator himself who checked whether the subgingival deposits had been removed or a second operator did the checking. 

Please include this information.

Our response: Thank you for your suggestion. The experienced operator verified the results of subgingival instrumentation himself based on a high ethical approach.

Revised text: not applicable.

Reviewer’s concern #13: Page 5, lines 233 and 234

“…Group C (negative control), a placebo gel, with similar aspect and consistency as the products used in Groups A & B…”

Here two points must be considered.

First, the authors should check the definitions of the negative control group and placebo control group; they are different and it seems to me the authors use the terms as synonyms:

-       Negative control group: In this type of control group, the participants are not given a treatment. The experimental group can then be compared to the group that did not experience any change or results.

-       Placebo control group: This type of control group receives a placebo treatment that they believe will have an effect. This control group allows researchers to examine the impact of the placebo effect and how the experimental treatment compared to the placebo treatment.

Second, in lines 201 and 202 “…Due to the particular physical nature and package of the products, only the patients and the evaluator were blinded…”

Therefore, if the two active gels were different, shouldn't the present study have two placebos? With one placebo simulating each of the different gels.

According to the description of the materials and methods this care was not taken. Therefore, the authors should justify this point, since it may generate a "nocebo effect" inserting biases in the study.

First - our response: Thank you for the suggestion. We deleted the term “negative control” and reformulated the sentence.

Revised text: “…Group C (placebo), a placebo gel, was applied into the periodontal pocket in the most apical portion.

Second – our response: since the patient received only one gel and had no idea about the efficacy of any of the gels, from his/her point of view all three gels could have been either active or inactive, despite the differences in appearance. Besides, the placebo gel was definitely innocuous, and there was no warning against potentially negative effects of the products, so a “nocebo effect” was out of question. Since the operator was not blinded, there was no need for two placebo gels.

Revised text: not applicable.

Reviewer’s concern #14:

Statistical Analysis

Page 5, line 248

“…are needed to detect a significant mean difference of 1 mm in…”

What were the criteria used by the authors to define that 1 mm would be the minimum difference to be considered clinically relevant in reducing clinical pocket depth? A 1mm reduction in clinical pocket depth can be achieved with supragingival instrumentation and hygiene orientation alone. 

Please clarify this point.

Our response: We are not sure if we understood exactly your concern. On one side there is the improvement due to subgingival instrumentation (see, for example, Cobb 2003 or other reviews), on the other side is the amount of the difference between the groups due to adjunctive treatments. The sample size calculation was based on earlier reports: Goodson JM et al. 2012, Griffiths et al. 2011, Guerrero et al. 2005, and more (considering that the standard deviation in previous studies was 1 mm). Power analysis calculations were performed before the study was initiated. To achieve 80% power and detect mean differences of the clinical parameters between groups, a significant mean difference of 1 mm (assuming a common standard deviation of 1 mm and given significance level α=0.05) was needed.

  1. Goodson JM, Haffajee AD, Socransky SS, Kent R, Teles R, Hasturk H, Bogren A, Van Dyke T, Wennstrom J, Lindhe J. Control of periodontal infections: A randomized controlled trial I. The primary outcome attachment gain and pocket depth reduction at treated sites. J Clin Periodontol 2012; 39: 526–536. doi: 10.1111/j.1600-051X.2012.01870.x.
  2. Griffiths, G.S., Ayob, R., Guerrero, A., Nibali, L., Suvan, J., Moles, D.R. and Tonetti, M.S. (2011), Amoxicillin and metronidazole as an adjunctive treatment in generalized aggressive periodontitis at initial therapy or re-treatment: a randomized controlled clinical trial. Journal of Clinical Periodontology, 38: 43-49. https://doi.org/10.1111/j.1600-051X.2010.01632.x
  3. Guerrero, A., Griffiths, G.S., Nibali, L., Suvan, J., Moles, D.R., Laurell, L. and Tonetti, M.S. (2005), Adjunctive benefits of systemic amoxicillin and metronidazole in non-surgical treatment of generalized aggressive periodontitis: a randomized placebo-controlled clinical trial. Journal of Clinical Periodontology, 32: 1096-1107. https://doi.org/10.1111/j.1600-051X.2005.00814.x
  4. Boia, S., Boariu, M., Baderca, F., Rusu, D., Muntean, D., Horhat, F. ... Stratul, Åž. (2019). Clinical, microbiological and oxidative stress evaluation of periodontitis patients treated with two regimens of systemic antibiotics, adjunctive to non-surgical therapy: A placebo-controlled randomized clinical trial. Experimental and Therapeutic Medicine, 18, 5001-5015. https://doi.org/10.3892/etm.2019.7856

Revised text: not applicable.

Reviewer’s concern #15: Page 6, lines 256 to 259

“…using Kruskal–Wallis tests, with post–hoc Mann–Whitney pairwise tests, as necessary. Proportions were compared by chi square tests. Assessment of intragroup differences between successive time points for quantitative variables was performed using Friedman tests, with subsequent Wilcoxon signed-rank tests for pairwise…”

The authors conducted the data analysis only with non-parametric models (Kruskal-Wallis, Mann-Whitney, Friedman and Wilcoxon); however, according to Cohen (2001): “For the analysis of dental indices, even those that are not normally distributed, parametric methods are generally valid, usually more powerful, and always more versatile than non-parametric alternatives. Routine rejection of parametric tests because data are non-normal ignores the documented robustness of these methods under common design features of dental research. Rejection of parametric tests because data are ordinal is not required by statistical theory. Selection of appropriate statistical methods should not be dictated by measurement-scale typology but should be guided by careful evaluation of the scientific meaning and the empirical value of those measurements.”

Reference: Cohen ME. Analysis of ordinal dental data: evaluation of conflicting recommendations. J Dent Res. 2001 Jan;80(1):309-13. doi: 10.1177/00220345010800010301. PMID: 11269721.

Therefore, as a suggestion (if indeed) I recommend that the authors think of alternatives for analyzing the data so that the results can be confirmed.

Our response: Although we are aware of the ongoing debate regarding the use of parametric methods for non-normal data, given the limited sample size available we consider it is more appropriate to go with a nonparametric approach. Additionally, in the case of ordinal data (detection scores of bacterial species), due to the manner in which the scores are assigned, it would be improper to treat these measurements as quantitative (interval-like).

Revised text: not applicable.

Results

Reviewer’s concern #15: Page 6, lines 271 and 272

“No side effects or negative impacts connected to the treatment methods were found during the study period.”

I recommend that the authors include more detail in this paragraph. The authors should consider reporting pain after subgingival instrumentation and dentinal hypersensitivity.

Our response: Thank you for your suggestion. However, pain as well as hypersensitivity were not variables of the study protocol and were not reported by the patients. According to the producer, after treatment with Ligosan Slow Release, the side effects are common with those of the standard periodontitis treatment. Uncommon side effects could have been swelling of the gingiva (periodontal abscess), chewing-gum like taste when the gel comes out of the gingival pocket. Again, none of them was noticed. As it has been proven that the application of Ligosan Slow Release only leads to very low doxycycline plasma concentrations, the occurrence of systemic side effects is very unlikely. There were no general disorders and administration site conditions, like hypersensitivity reactions, urticaria, angioneurotic oedema, anaphylaxis, anaphylactic purpura. For Gelcide, the producer mentions that side effects are eliminated because of local application at site of infection. No side effect was noticed for Gelcide, as well. The placebo gel elicited no adverse events or side effects, as well.

Revised text: No side effects or negative impacts like pain or dentinal hypersensitivity directly connected to the applied gels were reported.  

Reviewer’s concern #17: Page 8, Table 1

In the probing pocket depth (PPD) the authors need to include the result of the statistical test to confirm that the distribution of the probing pocket depth (PPD) in millimeters confirms the percentage distribution.

Our response: The data was included in the Table 1, as suggested. Thank you.

Revised text: not applicable, please see Table 1. 

Reviewer’s concern #18: Page 8, Figure 3

The authors need to make it clear in the legend of figure 3 what the graph represents: mean and standard deviation or median and interquartile range. I recommend that all dots be included in the figure, not just the outliers.

Please check, Drummond & Vowler (2011): “The present article encourages authors to present data clearly, preferably as a dot plot, so that the distribution of the values can be recognized”.

The p-values presented have 6 decimal places; I recommend a maximum of 4 decimal places.

Change the information about the groups: instead of using Group A, I recommend changing the treatment used.

Reference:

Drummond GB, Vowler SL. Show the data, don't conceal them. Br J Pharmacol. 2011 May;163(2):208-10. doi: 10.1111/j.1476-5381.2011.01251.x. PMID: 21501140.

Our response: The data was included in the figure, the number of decimal places for p-values was limited to 3 and the group names were changed as suggested.

Revised text: The evolution of the PPD between successive timepoints in the three groups (numbers in brackets represent p-values of Wilcoxon tests for pairwise intra-group comparisons) boxplot depicts median and interquartile range, large stars represent outliers; data points displayed as superimposed dot plot.

Reviewer’s concern #19: Page 8, lines 309 and 310

“However, despite 309 these modifications of the parameters, the variations among the groups never reached statistical significance…”

This reviewer has a slightly different opinion than the authors regarding the statement. The impression I get is that the authors are justifying the fact that the results did not reach significant differences ("the sacred p-value"). I simply state the following "the groups had similar results".

Our response: We edited the text as you suggested.

Revised text: “The evolution of PPD follows a similar dynamic in the 3 groups: a pronounced decrease after the first 3 months, followed by a slight increase at 6 months. However, the groups had similar results.”

Reviewer’s concern #20: Page 9, Figure 4

Check the same considerations made for figure 3.

Our response: done

Revised text: The evolution of the REC between successive timepoints in the three groups (numbers in brackets represent p-values of Wilcoxon tests for pairwise intragroup comparisons; p-values>0.05 are not displayed), boxplot depicts median and interquartile range, large stars represent outliers; data points displayed as superimposed dot plot.

Reviewer’s concern #21: Page 10, Figure 5

Check the same considerations made for figure 3.

Our response: done

Revised text: The evolution of the CAL between successive timepoints in the three groups (numbers in brackets represent p-values of Wilcoxon tests for pairwise intragroup comparisons; p-values>0.05 are not displayed), boxplot depicts median and interquartile range, large stars represent outliers; data points displayed as superimposed dot plot.

Reviewer’s concern #22: Page 10, line 304

“…the values in group B being apparently…”

This statement is not adequate, since the results had no significant differences. We cannot state that apparently, the results are lower. Since the test had no significant difference, we can only state that they were similar.

Our response: The text was corrected accordingly.

Revised text: Comparisons of FMPS measurements show slight differences between the groups at 3 months; however, subsequent pairwise comparisons using Mann-Whitney tests do not indicate these differences to be statistically significant (p>0.05 in all cases, after applying the correction for multiple comparisons).

Reviewer’s concern #23: Page 10, Figure 6

Check the same considerations made for figure 3.

Our response: done

Revised text: The evolution of the FMPS between successive timepoints in the three groups (numbers in brackets represent p-values of Wilcoxon tests for pairwise intragroup comparisons; p-values>0.05 are not displayed), boxplot depicts median and interquartile range, large stars represent outliers; data points displayed as superimposed dot plot.

Reviewer’s concern #24: Page 11, Figure 7

Check the same considerations made for figure 3.

Our response: done

Revised text: The evolution of the FMBS between successive timepoints in the three groups (numbers in brackets represent p-values of Wilcoxon tests for pairwise intragroup comparisons; p-values>0.05 not displayed), boxplot depicts median and interquartile range, large stars represent outliers; data points displayed as superimposed dot plot.

Reviewer’s concern #25: Page 11, Table 2

Please explain the missing values in Table 2. Information should be added at the bottom of the table.

Our response: The missing values indicate frequencies=0. They have been replaced with zeroes.  As for the issue of Table 2, unfortunately Reviewer No.1 insisted for its elimination as unnecessary and for presenting these data in the text, focusing on those species that changed in statistically significant way at 6 moths. This is what we did in the text below.

Revised text:

3.2. Results of microbiological tests

The microbiological results (A. actinomycetemcomitans, P. gingivalis, P. intermedia, T. forsythia, T. denticola) were without statistical significance between all three groups at baseline and after 6 months (p = 0.190 – 0.859, respectively). The statistical analysis within the groups resulted in significantly lower detection scores for P.g., P.i., T.f., and T.d. after six months for all groups. The p - values were between 0.007 and 0.029. There was no change of the bacterial counts of A. actinomycetemcomitans in the control group after six months. Relating to A.a., in the control group particularly, the numbers observed at baseline, are comparable with those after 6 months. Corresponding to Kruskal-Wallis tests for intergroup comparisons of pathogen.

Discussion

Reviewer’s concern #26: Page 13, lines 434 and 435

“The present treatment protocol included one round of application of piperacillin plus tazobactam, in periodontal pockets evaluated at 3-6 months the application.…”

The authors applied the gels in a single round. This point should be further explored in the discussion. If the study had performed a second instrumentation session with antimicrobial gel application at 3 months could the results be different?

Our response: Thank you, this is a very interesting discussion. At the time of submitting this manuscript, no other clinical trial addressed the issue of multiple applications of piperacillin plus tazobactam in periodontal pockets. Therefore, we could not speculate whether the results could have been different. On the other hand, a more recent systematic review and meta-analysis on the adjunctive effect of locally delivered antimicrobials in periodontitis therapy noted: “The number of applications varied among products and study protocols, being the most frequent just one application, in 34 study groups; two applications were performed in 10 study groups and more than two in five. In six study groups, one initial application was performed; while a second (three studies) or a third one (three studies) was decided based on the dislodging on the first application or on the presence of pockets. When more than one application was scheduled, the protocols were highly heterogeneous” (Herrera et al. 2020). The same applies for controlled-release doxycycline: “Although short-term (3 months) beneficial effects on clinical parameters were demonstrated with the adjunctive use of locally delivered controlled-release doxycycline in periodontal maintenance patients, repeated application once annually had no long-term clinical and microbiologic effects above and beyond those observed with subgingival mechanical debridement alone”. (Bogren, Anna; Teles, Ricardo P.; Torresyap, Gay; Haffajee, Anne D.; Socransky, Sigmund S.; Wennström, Jan L.  (2008). Locally Delivered Doxycycline During Supportive Periodontal Therapy: A 3-Year Study. Journal of Periodontology, 79(5), 827–835.         doi:10.1902/jop.2008.070515).

Revised text: not applicable.

Reviewer’s concern #27: Page 13, lines 434 and 435

The authors did not adequately explain both the strengths and weaknesses of the observations should be discussed."

Our response: Lines 434 and 435 of the original manuscript seem not to be related to your concern. The following strengths and weaknesses of the observations were added and commented: The major strength of our study is the fact that this is the first study comparing piperacillin plus tazobactam gel with doxycycline and a placebo gel as an adjunct to subgingival instrumentation. The weaknesses are presented already as limitations in the last paragraph of the Discussion section.

Revised text: “The major strength of this study is the fact that this is the first study comparing piperacillin plus tazobactam gel with doxycycline and a placebo gel as an adjunct to subgingival instrumentation. The limitations of this study include the unknown duration of the effect of the adjunctive topical administration of antibiotics and the relatively short follow-up period”.

Reviewer’s concern #28: Questions regarding the costs (dollar or euro) of antimicrobial gels for use in periodontal therapy should be discussed.

Our response: Thank you for your suggestion. This study was not intended to perform neither cost-effectiveness nor cost-minimization analyses of the discussed antimicrobial products. However, it happens that the cost per application of the test product (Gelcide) is approximately equal to that of the positive control (Ligosan), provided the manufacturer’ instructions are respected.

Revised text: the following phrase was added in the last paragraph of the Discussion section: “This finding is important when taking into consideration that the cost per a single application of Gelcide® is approximately equal to that of Ligosan®, if the manufacturer’s instructions are respected.”

Reviewer’s concern #29: The perceptions of the patient (patient-reported outcomes and patient-reported experience) and the operators should also be presented and discussed.

Check:

McGuire MK, Scheyer ET, Gwaltney C. Commentary: incorporating patient-reported outcomes in periodontal clinical trials. J Periodontol. 2014 Oct;85(10):1313-9. doi: 10.1902/jop.2014.130693.

Our response: The need to perform evidence-based clinical dentistry assessing patient satisfaction or other patient-related outcomes is unquestionable. However, such an analysis was not performed in our study, based on several facts: 1. The single local application mode of all products had nothing spectacular, or even special. 2. As in previous applications no patient complained about, or even noticed specific organoleptic characteristics of any of the products, we did not consider such an analysis in the present study. Nevertheless this could be a future research topic in a study on possible multiple applications.

Revised text: not applicable.

Reviewer’s concern #30: I recommend that the authors use NNT (number needed to treat) calculations to enhance determining the clinical relevance of this periodontal research finding. The authors could consider it as a secondary analysis of the data so that it can be discussed. I believe that the use of NNT can contribute to the clinical understanding of the results.

Check:

Greenstein G, Nunn ME. A method to enhance determining the clinical relevance of periodontal research data: number needed to treat (NNT). J Periodontol. 2004 Apr;75(4):620-4. doi: 10.1902/jop.2004.75.4.620.

Our response: At 6 months, we have a decrease in PPD with ≥ 2mm in group A 6/21, in group B 7/22 (31.8%) and group C la 6/21 (28.6%) respectively. Because group A and C are equal, the NNT is infinite. Since the results are similar between groups, the absolute risk reduction is very close to zero, in which case the NNT is not recommended to be calculated.

* Schechtman E. Odds ratio, relative risk, absolute risk reduction, and the number needed to treat--which of these should we use? Value Health. 2002 Sep-Oct;5(5):431-6. doi: 10.1046/J.1524-4733.2002.55150.x. PMID: 12201860.

Revised text: not applicable.

Reviewer’s concern #31

The biases of the study are not discussed.

Our response: Although a typical RoB2 assessment of risk of bias analysis (as per Higgins, J.P., Savović, J., Page, M.J., Elbers, R.G. and Sterne, J.A. (2019). Assessing risk of bias in a randomized trial. In Cochrane Handbook for Systematic Reviews of Interventions (eds J.P.T. Higgins, J. Thomas, J. Chandler, M. Cumpston, T. Li, M.J. Page and V.A. Welch). https://doi.org/10.1002/9781119536604.ch8) was not performed, all RoB2 requirements were addressed. In order to minimize the biases in the study, the following actions were taken: we reported the randomization and allocation methods in detail (selection bias); we presented the follow-up period and outcome data (attrition bias); we describe the blinding methods of participants and personnel as well (performance bias) and examiner blinding was performed strictly (detection bias); complete reporting of the data (reporting bias); some indirect sources of bias can be considered: the fact systemic diseases and medication were not reported, tooth types were not specified.

Revised text:

The text was added in the last paragraph of the Discussion section:

“Potential indirect biases of this study could be considered that systemic diseases and medication should have been considered, and tooth types for each patient were not specified.”

Reviewer’s concern #32

Conclusion:

The conclusion does not adequately reflect what the results showed.

The authors should elaborate a conclusion that better reflects the results, i.e., the groups had similar results, and according to the results we can use the subgingival instrumentation methods without the need to use antimicrobials, reducing the costs for the patient and the working time of the professional.

Our response: Thank you for your suggestion, we revised the conclusion accordingly.

Conclusion: The groups had similar results, and subgingival instrumentation can be executed  without adjunctive antimicrobials, reducing the costs for the patient and the working time/load of the professional.

Round 2

Reviewer 2 Report

Medicina

Manuscript: medicina-2142403

Title: A placebo-controlled trial to evaluate two locally delivered antibiotic gels (piperacillin plus tazobactam vs. doxycycline) in stage III–IV periodontitis patients.

Revision_1

I realized that the authors put a lot of effort into elaborating on my queries.

Just a very small suggestion:

Page 4, Line 170

“...immediate vicinity of the contact point with optimal pressure…”

I recommend the following.

“...immediate vicinity of the contact point with optimal pressure (20-25N)…”

Decision:

The authors performed or justified all the queries that were made.

In some statements, I have an opinion that disagrees with the authors' arguments. However, this does not mean that I am right in my opinion, since these differences of opinion are present in the literature and are part of science.

Author Response

Page 4, Line 170

“...immediate vicinity of the contact point with optimal pressure…”

 I recommend the following.

“...immediate vicinity of the contact point with optimal pressure (20-25N)…”

Thank you for the suggestion. The text was revised accordingly